# A Pix2Pix Architecture for Complete Offline Handwritten Text Normalization

**DOI:** 10.3390/s24123892

**Published:** 2024-06-16

**Authors:** Alvaro Barreiro-Garrido, Victoria Ruiz-Parrado, A. Belen Moreno, Jose F. Velez

**Affiliations:** Higher Technical School of Computer Engineering, Universidad Rey Juan Carlos, c/Tulipan sn, Mostoles, 28922 Madrid, Spain; alvaro.barreiro@urjc.es (A.B.-G.); victoria.ruiz.parrado@urjc.es (V.R.-P.); belen.moreno@urjc.es (A.B.M.)

**Keywords:** offline handwriting, scanned text preprocessing, image normalization, IAM dataset, GANs, pix2pix, deep learning

## Abstract

In the realm of offline handwritten text recognition, numerous normalization algorithms have been developed over the years to serve as preprocessing steps prior to applying automatic recognition models to handwritten text scanned images. These algorithms have demonstrated effectiveness in enhancing the overall performance of recognition architectures. However, many of these methods rely heavily on heuristic strategies that are not seamlessly integrated with the recognition architecture itself. This paper introduces the use of a Pix2Pix trainable model, a specific type of conditional generative adversarial network, as the method to normalize handwritten text images. Also, this algorithm can be seamlessly integrated as the initial stage of any deep learning architecture designed for handwritten recognition tasks. All of this facilitates training the normalization and recognition components as a unified whole, while still maintaining some interpretability of each module. Our proposed normalization approach learns from a blend of heuristic transformations applied to text images, aiming to mitigate the impact of intra-personal handwriting variability among different writers. As a result, it achieves slope and slant normalizations, alongside other conventional preprocessing objectives, such as normalizing the size of text ascenders and descenders. We will demonstrate that the proposed architecture replicates, and in certain cases surpasses, the results of a widely used heuristic algorithm across two metrics and when integrated as the first step of a deep recognition architecture.

## 1. Introduction

Despite recent advances driven by deep learning, the reliable recognition of handwritten text, captured from either cameras or scanners, is still part of the future. Therefore, handwriting could become a significant means of communication with computers, given factors such as privacy, the widespread use of paper, and its importance in early human learning processes. Many attempts have been made over the years to develop precise HTR systems. Prior to 2013, many solutions were based on Hidden Markov Models (HMMs) as prevailing architectures [1,2,3]. From 2013 onwards, however, deep learning models have been considered as the standard methodology for offline text recognition.

The intra-personal variability between writers has always been a limiting factor in the capabilities of offline handwritten text recognition (HTR) systems, as each person has their own handwriting style. Also, several other factors such as the thickness of the strokes due to the type of pen used may have an impact on the readability of a given text. The improvement of this readability may boost subsequent recognition results. In order to overcome these issues, a previous text normalization stage has been established as common practice for most HTR algorithms. This preprocessing step is in consonance with the classic methodology in many types of computer vision problems: the images have to be normalized before the subsequent recognition models are applied [4,5].

In this context, the aim of the normalization step is to reduce the intrinsic variability within the handwritten text images before passing them to an HTR model. This preprocessing task may include several substeps, such as text binarization, image noise removal, slope correction, slant correction, and normalization of ascenders and descenders [6,7,8]. Classical solutions were based on heuristics algorithms [9,10]. Some authors omitted this normalization process [11,12] with mixed results, but those who have applied normalization have typically obtained as good and usually better results with similar recognition models [5,6,13,14,15,16]. In recent years, many authors have researched preprocessing tasks, but most of them were focused only on one of the substeps mentioned above.

Generative Adversarial Neural Networks (GANs) are popular deep learning models that are able to generate high-quality images [17]. In particular, Pix2Pix [18] models are conditional GANs that are able to face what is called paired image-to-image translation. Some works within the literature deal with the use of GAN for image normalization purposes [19]. In the area of automatic document analysis, GAN models have been used to delete background and printed text elements [20], denoise [5], deblur [21] or binarize [22] the images, among others. Some GAN models also normalize the text width with erosion and dilation transformations [23] or try to remove the noise present in document images [24]. However, note that none of these works are focused on the complete normalization process. In this sense, this work presents a new way of action, in which the normalization process is performed by a network that eventually could be trained in an end-to-end configuration that would combine normalization and recognition within a single scheme.

This paper presents a novel handwritten text normalization algorithm based on a Pix2Pix model trained from scratch. The contributions of this paper can be summarized as follows:The proposed model reduces the intra-personal variabilities present in the handwritten text of subjects. As far as we know, the complete handwriting text normalization problem with all its substages has not been addressed as a whole by any deep learning model. Moreover, the novelty of our proposal relies on the application of a Pix2Pix network for normalizing handwritten text while preserving the characters and the legibility of the word, rather than for generative purposes.The adjustment made to the original Pix2Pix architecture to tackle this particular problem. This adaptation involves a decrease in the number of layers of the U-Net generator and therefore the subsequent reduction in the number of parameters that is necessary to train the model.A comparison between the proposed solution and an ad hoc heuristic procedure.

The rest of this paper is organized as follows: Section 2 discusses related works, Section 3 covers the dataset, the preprocessing steps, and the architecture used for normalization. Section 4 describes the experiments and results, whereas Section 5 concludes with the main findings and future directions.

## 2. Related Works

Different authors have proposed and applied several heuristic normalization strategies over the years in order to improve their HTR models.

Most initial works followed a pattern that typically included similar stages to accomplish the handwritten preprocessing task, namely text binarization, image noise removal, slope correction, slant correction, and normalization of ascenders and descenders. For such preprocessing, image-based techniques were focused on finding automatic thresholds to binarize text and remove noise [25]; on using either Hough transform [26] or horizontal projections [27] to correct the slope; on pixel projections [28], pixel counting [29], or other techniques to calculate the text-cursive leaning and correct the slant; or on baseline detection to normalize the size of ascenders and descenders [30].

Some authors develop complete systems to normalize the images. Those systems are based on heuristic methods. Sueiras et al. [31] propose a combination of heuristic methods to normalize the images. They try to fix most of the common problems regarding HTR, such as noise reduction, binarization, image inversion, slant correction, baseline detection, slope correction, and normalization of ascenders, descenders, and resizing.

More recent works use deep learning techniques in order to achieve automatic preprocessing. Most of them are mainly focused on image binarization and document image enhancement. For example, Akbari et al. [32] and Calvo-Zaragoza et al. [22] applied Convolutional Neural Networks (CNNs), long short-term memory (LSTM), and autoencoders to address these issues.

Other works that also tackle preprocessing are mainly focused on GAN models.

Sadri et al. [33] use a CycleGAN architecture to enhance document images. Another work of this research group uses GAN models for data augmentation [34].

Gonwirat and Surinta [24] handle image denoising via GAN models. They developed both a GAN devoted only to image enhancement and a combination of GAN and CNN models for the enhancement and recognition of handwritten characters.

Souibgui and Kessentini [21] focused their work on the enhancement of document images, this time through the use of conditional GAN models (cGANs). This improvement aimed to remove warping, blurring, watermarks, etc. Their work was extended to train a GAN to enhance the images jointly with a Convolutional Recurrent Neural Network (CRNN) devoted to handwritten text recognition [20]. In this work, the GAN architecture was composed of a U-Net generator and a Fully Convolutional Network discriminator.

Suh et al. [35] managed to remove most of the document image background, including text on the reverse page, stamps, lines, etc. The architecture selected in the first stage is a Pix2Pix GAN, which removes background and color information. Specifically, it is composed of four U-Net neural networks as four color-independent generators and a PatchGAN [18] variation as discriminator. Right after that, in the second stage, local and global binary transformations were applied to improve the first-stage images.

Kang et al. [23] pretrained and combined several U-Nets to deal with specific preprocessing tasks such as dilation, erosion, histogram-equalization, Canny-edge, and Otsu binarization. Their system is specially designed to handle small datasets.

Zhao et al. [36] use conditional GANs to binarize images. More specifically, a first conditional GAN is trained to generate images, whereas a subsequent cascade of conditional GANs is used to compose the image by taking into account multi-scale features.

Kumar et al. [37] describe an unsupervised model which does not need paired images for binarization. These authors create augmented texture images through a GAN model, pseudo-pairing them with the original ones and train a discriminator to differentiate them.

The problem of mimicking the writing style of authors has been investigated by Davis et al. [38]. These authors proposed a model that combined both GANs and autoencoders to capture global style variations of the writers, such as size or slant.

## 3. Materials and Methods

This section provides a brief review of the handwritten text image databases used to train the models. Then, we explain the heuristic normalization scheme that will be performed. The normalized images of each database will be the ground truth for our trainable models, which will be outlined in the last subsection.

### 3.1. Handwriting Datasets Used

In this work, we utilized three distinct handwriting datasets: IAM, Bentham, and Osborne.

IAM [39] is an English dataset comprising scanned forms of handwritten text. The forms are segmented into text lines, sentences, and words. The latest version of IAM consists of 115,320 isolated and labeled words distributed over 13,353 text lines from 657 different writers. This was achieved by scanning 1539 pages of text at a 300 DPI resolution. The scanned images were saved in PNG format with 256 gray levels. The IAM partitions used throughout this work are made up of 47,926 words for the training set, 7558 for validation, and a total amount of 20,292 words for the test set.

Bentham is also an English handwritten dataset [40], but in this case, it has been obtained from historical documents written by a single author: the philosopher Jeremy Bentham (1748–1832). The database comprises 11,473 text lines partitioned into 9198 lines for training, 1415 for validation, and 860 for testing. As this database does not provide segmentation at word level, we have used pseudo-words created by cropping the lines at a fixed step size, with the width being six times the height.

The Osborne dataset is based on the Osborne Archive [41], which includes a collection of handwritten letters exchanged between notable figures from various periods, spanning from 1830 to 1883. The corpus of this dataset is made of 200 RGB-digitized and transcribed documents from the original archive at a 400 DPI resolution. The Osborne dataset is a notable example of a multi-language historical handwritten text database, featuring primarily Spanish texts, with some English and Italian ones. This word image dataset, the smallest of the three, is divided into 7149 words for training, 288 for validation, and 194 images for testing. As this dataset does not provide information at the line level, we have used the adjacent words as context when necessary.

### 3.2. Normalization Scheme

The heuristic normalization performed on the handwritten images follows the procedure developed by Sueiras et al. [14] and will serve as ground truth for our trainable model.

The following preprocessing steps have been applied to the text line images, rather than directly to the word images. In doing so, some of the selected algorithms severely increase their robustness. This is especially determining in the case of the slope removal algorithm. The normalization method is described in Algorithm 1. The next subsections will describe the normalization steps in detail.
**Algorithm 1** Heuristic normalization**function**Normalize(imgDatabase)   ouputDatabase = [ ]   **for** lineImg∈imgDatabase **do**:         base = detectBaseLine(lineImg)         lineImg = slopeCorrection(lineImg, base)         lineImg = slantCorrection(lineImg)         b,u = detectBaseAndUpperLine(lineImg)         lineImg = normAscendersDescenders(lineImg,*b*,*u*)         wordList = cropWords(lineImg)         **for** wordImg∈wordList **do**:                  ouputDatabase← resizeWordImage(wordImg)         **end for**   **end for**   **return** ouputDatabase**end function**

#### 3.2.1. Image Inversion

We inverted the images with respect to the maximum value of 255 so that the handwritten text strokes have the highest values. On the contrary, background pixels are close to 0. It has been observed in several experiments that this helps CNN networks to produce slightly better results [42].

#### 3.2.2. Slant Correction

Handwritten text images were deslanted using the algorithm provided by [43]. A threshold value of 128 was used to distinguish between black and white pixels. If a pixel is considered as a boundary one (a black pixel with a white one on its right), then the values of the top three adjacent pixels with respect to that position are analyzed. In doing so, the algorithm accumulates evidence about the local inclination of the text strokes. Once the slant angle α is estimated, an affine transformation is applied to correct it.

#### 3.2.3. Baseline Detection and Slope Correction

To correct the slope of the text lines, we have aimed for a previous identification of the base and upper lines, using an approach similar to [30]. Baseline detection of each text line was carried out by applying a Random Sample Consensus (RANSAC) regression [44] over the lowest points of each column of pixels (with values above a threshold equal to 20). The baseline position with respect to a horizontal line yields an estimate of the slope angle, that will be corrected by an affine transformation. The upper line is obtained afterwards, with the RANSAC procedure being repeated over the highest points of the line image that fulfill the same criterion (although in this case, a threshold of 128 was used).

#### 3.2.4. Normalization of Ascenders and Descenders

After slope correction, base and upper lines are recalculated with RANSAC, assuming that they are now straight horizontal lines. The image gap between them delimits the main body of text, leaving ascenders and descenders as the horizontal top and bottom ends, respectively. These regions are rescaled to half their original size and concatenated again with the core region.

#### 3.2.5. Resizing

Finally, we extract the word images from this line-preprocessing procedure. This has been achieved using the *x*, *y*, *w*, *h* bounding box coordinates of each word within a given line image. Next, their sizes are fixed to a 48×288 resolution, preserving the aspect ratio of the contained handwritten words.

### 3.3. The Pix2Pix Architecture

Our Pix2Pix model aims to normalize handwritten text by mapping non-normalized images to their normalized counterparts. The model receives two sets of images: one normalized according to Section 3.2 (ground truth), and a second raw version of the very same set. The Pix2Pix network normalizes the non-normalized images while preserving legibility, using a generator and a discriminator like other GAN architectures.

#### 3.3.1. The Pix2Pix Generator: U-Net

The first component of a Pix2Pix model is the U-Net generator [45]. Figure 1 displays a scheme of our U-Net architecture. The input (top-left corner) of the U-Net generator would be the raw, non-normalized handwritten word images. U-Nets usually work as an encoder–decoder model. Thus, during the decoder expanding path (right side of Figure 1), the U-Net produces as output the normalized version of the input image. In order to avoid losing information during the encoding stage, the model introduces skip connections (gray arrows in Figure 1).

Notice that both the encoding and the decoding paths have been reduced to 4 layers (in contrast with the 8 layers of the original Pix2Pix scheme [18]). This is quite reasonable since our training images only have a resolution of 48×288. Nevertheless, the latter implies a significant reduction in the trainable parameters and therefore a considerable decrease in the training time as well.

#### 3.3.2. The Pix2Pix Discriminator: PatchGAN

The Pix2Pix discriminator, PatchGAN, outputs a matrix of classifications. In our case, each value of this matrix scores the normalization performance of the generator with respect to the raw, non-normalized, image. It takes as input a combination of the raw images and either those normalized using the heuristic preprocessing scheme (Section 3.2), or those provided by the generator. By sliding its field of view across all the patches of the input image, the PatchGAN will then give feedback on how good the normalization process has been for each of these patches. The closer the score to 1, the better the preprocessing of that region. Ideally, the PatchGAN will try to output a matrix of zeros for an image coming from the generator and a matrix of ones for an image preprocessed using the heuristic. However, as the generator improves in its normalization task, it becomes more difficult for the PatchGAN to determine if the given image comes from the generator or has been preprocessed with the scheme introduced in Section 3.2. The PatchGAN discriminator can also be expressed as a sequence of the following: a 2D convolution with 8 filters and kernel 1; two contracting blocks (see Figure 1), the first of which lacking BatchNorm layers; a final 2D convolution with 32 filters and kernel 1. This outputs a final (1×12×48) tensor containing the normalization scores.

#### 3.3.3. Loss Function and L1 Distance

Originally, GANs were generative models, *G*, that learn to map from a random noise vector *z* to an output image *y*, G:z→y [17]. However, in conditional GANs, they learn the mapping from an observed image *x* (i.e., our raw handwritten text images) and some random noise vector, *z* to *y*, G:{x,z}→y. Our Pix2Pix U-Net does not even need that random noise vector, which yields: G:x→y. As stated above, the U-Net generator, which will be the map *G*, will be trained to output normalized images that cannot be distinguished from those produced using the heuristic procedure stated in Section 3.2. Quite the contrary, the PatchGAN discriminator, *D*, will try to measure the quality of the normalization by telling whether the image produced came from the generator or from the heuristic preprocessing scheme (ground truth images) given by [14]. The typical loss function for a conditional GAN is defined as Equation (Equation 1) [18]:(1)$$L_{cGAN}\left(G,D\right)=E_{x,y}\left[logD\left(x,y\right)\right]+E_{x}[log\left(1-D\left(x,G\left(x\right)\right)\right]$$

Bearing in mind that the goal of the generator is to produce normalized handwritten text images as close as possible to the normalization provided by the ground truth set, we introduce an L1 distance term to the loss function.

The L1 distance measures the similarity between generated and ground truth images. It is a reconstruction term with a weight parameter $\lambda_{recon}$ used to control its impact on the global loss function. The L1 term is defined as
(2)$$L_{L1}\left(G\right)={E_{x,y}[∥y-G\left(x\right)∥}_{1}]$$

Hence, the final expression of the loss function is
(3)$$L\left(G,D\right)=L_{cGAN}\left(G,D\right)+\lambda_{recon}L_{L1}\left(G\right)$$

## 4. Experiments and Results

The very next step consists of computing the accuracy and effectiveness of our Pix2Pix model. The workflow followed in each experiment of this section (including train and test stages) has been summarized in Figure 2.

The first metric selected to estimate the accuracy of the normalization process for this workflow is the average Distance Per Pixel (DPP hereafter).

Once trained (train region in Figure 2), the Pix2Pix model will perform normalization on the unseen test set images (test region in Figure 2). Then, we will evaluate the pixel-to-pixel L1 distance given by Equation (Equation 2) between the Pix2Pix output images and a certain set of ground truth counterparts. Let *i* be an image of the test image set *I* with range in [0,255], and let $i_{n}$ be the *n*-th pixel of *i*. We define $N_{1}\left(i\right)$ and $N_{2}\left(i\right)$ as two different normalizations that we want to compare after being applied on *i*. Then, for an individual image, the DPP is calculated as
(4)$$DPP\left(i\right)=\frac{1}{255\cdot h\cdot w}\sum_{n=1}^{h\cdot w}{∥N_{1}\left(i_{n}\right)-N_{2}\left(i_{n}\right)∥}_{1}$$
where *h* and *w* correspond to the height and width of the images.

In order to reinforce the reliability of our analysis, we computed a second metric, the *Structural Similarity Index* [46] (SSIM hereafter), to measure and compare the effectiveness of the aforementioned normalization methods. This parameter estimates the similarity between two images by mimicking the human visual perception, and it is given by the following:(5)$$SSIM\left(i\right)=\frac{\left(2\mu_{N_{1}\left(i\right)}\mu_{N_{2}\left(i\right)}+C_{1}\right)\left(2\sigma_{N_{1}\left(i\right)N_{2}\left(i\right)}+C_{2}\right)}{\left(\mu_{N_{1}\left(i\right)}^{2}+\mu_{N_{2}\left(i\right)}^{2}+C_{1}\right)\left(\sigma_{N_{1}\left(i\right)}^{2}+\sigma_{N_{2}\left(i\right)}^{2}+C_{2}\right)}$$
where $\mu_{N_{j}}$, $\sigma_{N_{j}}$, and $\sigma_{N_{j}N_{k}}$ are the mean, variance, and covariance of the image *i* normalized through $N_{j}$, respectively. For images in byte integer representation, C1=255×0.01 and C2=255×0.03, as specified by the authors of [46]. In summary, and given two images, SSIM outputs values in a range between −1 and 1 depending on how close those images are between them in terms of likeness. Hence, totally different images yield a value of −1, whereas the higher the similarity between them, the closer the outcome is to 1. Therefore, SSIM explores global similarities between images that have been normalized through two different processes, whereas DPP is constrained to a pixel-level analysis. Once obtained using Equations (Equation 4) and (Equation 5), DPP and SSIM values are averaged over the whole considered dataset.

### 4.1. Summary of Experiments

We conducted four distinct experiments using the three different raw datasets. These experiments aimed to determine whether our proposed method can replicate the heuristic transformations performed by Sueiras et al.’s normalization algorithm, which serves as our benchmark. For each experiment, we applied either our normalization algorithm or the heuristic one to different variants of the raw datasets.

The experiments on the various datasets were conducted using the presented Pix2Pix architecture, which was trained with the respective training portion of each dataset. However, due to the smaller size of the Osborne dataset and the limited diversity of the Bentham dataset (a single author), the training for these datasets started from a model pretrained on the IAM dataset.

Also, to measure the impact of the reconstruction term on the final normalization performance, we trained with values of $\lambda_{recon}=50,\;\lambda_{recon}=500$, and $\lambda_{recon}=5000$ for Equation (Equation 3). Thus, we tested three different variants of the network for each of the three datasets.

The main aim of each of these experiments may be summarized in the following items:Identifying a value of $\lambda_{recon}$ for achieving good results on the test partition of each dataset, using the reference heuristic algorithm outputs as ground truth.Assessing whether our proposal restores images from the test partition of each dataset when subjected to distortions, similar to the results obtained by the reference heuristic algorithm.Estimating how our proposal and the reference heuristic algorithm approximate manual ad hoc normalization performed on a subset of the IAM test set.Conducting a preliminary comparison between the text recognition results obtained when using our Pix2Pix normalization approach as a step previous to recognition and those achieved using the reference heuristic normalization.

#### 4.1.1. Experiment 1—Heuristic Replication Measurement

The first experiment consists of testing the extent to which our proposal is able to recreate the heuristic normalization, as this procedure acted as ground truth during our model training time. To test this assertion, we ran our three variants of the proposed Pix2Pix model on the raw test sets of the three datasets. Once the output normalized images have been obtained, we compare them with those normalized using the scheme presented by Sueiras et al. [14]. Regarding the corresponding results, Table 1 provides the output metrics of the first experiment. As it can be observed, there are no clear advantages when using different values for λ. In particular, the Pix2Pix variant with $\lambda_{recon}=5000$ (values in bold) gave the best results in terms of both DPP and SSIM for all tested databases. For example, for the IAM database, the model outputs a DPP value of 0.042, which means that, on average, each output pixel differs only by 4.2% with respect to its heuristic ground truth counterpart.

#### 4.1.2. Experiment 2—Distorted Datasets

We also provide an alternative version of the databases (IAM, Bentham, and Osborne) to test the accuracy of the normalization procedures when being applied to a different set of images. This alternative sets have been obtained by applying several mild-deforming transformations on the original raw test images. Thus, we will refer to these data as the *distorted datasets* henceforth. To obtain this modified versions, we have performed a pipeline of transformations on the raw text images. These transformations are made up of several steps and each of them is applied according to a certain probability. Hence, for each image, the pipeline of transformations involves the following steps: A 50% chance of undergoing an elastic distortion as the one proposed by Yousef and Bishop [47]. A 40% probability per image of being rotated with a random angle within the range of ±5 degrees. An 80% probability of adding a slant to the image, with random angles between ±0.5 degrees. Finally, a 30% probability of undergoing through either a 2×2 dilation or a 2×2 erosion. These probabilities have been determined by previous experiments.

These distorted images will hence be normalized by the three Pix2Pix reconstruction loss-variants described above and also by the reference heuristic method in order to compare the effectiveness of the normalization approaches when facing “unseen” handwritten text images. This time, however, each normalization was compared to itself being applied on the raw, non-distorted images. The results in terms of DPP and SSIM are presented in Table 2.

Even though the best outcomes were obtained by our model with $\lambda_{recon}=5000$, the obtained results were very similar for all the metrics. Therefore, we can conclude that both our proposal and the reference heuristic algorithm restore the test dataset images in a similar way, perhaps with some advantage for the proposed method.

#### 4.1.3. Experiment 3—Comparison with Manual Ad Hoc Normalization

In this third experiment, we randomly selected a sample of 100 images from the *raw IAM dataset*. These images were manually normalized by visual inspection, accounting especially for slope and slant corrections and preserving the aspect ratio. In addition, to fit the 48×288 resolution of our model, we ensured that the main body parts of these images were constrained to the central 24 pixels, whereas the other regions were left to ascenders and descenders. In doing so, we have generated an alternative ground truth set with which we are able to compare all the previously explained normalization algorithms. Figure 3 displays the raw version of one of these IAM test set images as well as its ad hoc normalized counterpart, whereas Table 3 encapsulates the values obtained for the computed metrics. In this case, we find that based on the DPP metric, our Pix2Pix proposal slightly surpasses the results of the heuristic algorithm for every considered λ parameter. The SSIM results, however, were very much alike for all the normalizations performed.

#### 4.1.4. Experiment 4—Examining Normalization Architecture in Deep Recognition

For our final experiment, we conducted a preliminary comparison between the results obtained from our normalization approach and those achieved using the heuristic normalization method proposed by Sueiras et al. and their own recognition model [14]. Sueiras et al. reported a Word Error Rate (WER) of 23.8% and a Character Error Rate (CER, based on Levenshtein distance) of 8.8% when applying their CNN+Seq2Seq LSTM-based attention model to the IAM test dataset with their heuristic normalization procedure, which we have adopted as the baseline for our work. After replacing their heuristic normalization with our Pix2Pix normalization method (see Figure 4), we observed a decrease in WER to 18.1% and in CER to 5.7% (see Table 4). This replacement occurred in two stages: first, we substituted the input layer of the CNN+Seq2Seq LSTM-based attention model with our pretrained Pix2Pix architecture; then, we trained the resulting model using the IAM training dataset, stopping if no further improvements in validation loss were observed for 100 epochs.

### 4.2. Discussion

Theoretically, the normalization stage quality should improve as the DPP decreases and SSIM increases, respectively. Following this rationale, the experiment with $\lambda_{recon}=5000$ provides some of the best results (see Figure 5). Nevertheless, a visual inspection of some of the network outcomes allows us to see whether this is true or not. In doing so, we will have a more qualitative and reliable intuition about the Pix2Pix normalization performance with respect to the “ideal” preprocessing of the ground truth images.

To illustrate the effectiveness of the model in a challenging case, we present the results of applying different normalization methods to a word image example, from the Osborne dataset, that exhibits overlapping text strokes (see Figure 6). It can be seen that our approach yields better results than the heuristic method and additionally reduces some of the noise present within the image.

We found that our Pix2Pix provides a good normalization with respect to heuristic preprocessing as long as the resizing factor (understood as the difference in the size of text between raw and heuristic images, relative to the entire image) is low. In such cases, most of the other features present in raw input images (slope, slant, low contrast between background and text) are properly corrected. Nevertheless, whenever this resizing of the text with respect to the raw input image becomes noticeable, the normalization performance of our model drops significantly.

Nonetheless, we mostly observe that for our conducted experiments, the trained Pix2Pix architecture effectively replicates the outcomes achieved by the reference heuristic algorithm for handwritten text normalization. These favorable results have facilitated the successful integration of this normalization stage into an end-to-end recognition architecture, enabling a comprehensive optimization of all model parameters.

## 5. Conclusions and Future Work

We present a Pix2Pix conditional GAN model to normalize handwritten text images. A normalized version of the IAM dataset, applying typical preprocessing routines on the original images, has served as the pretrained ground truth for the presented model. Also, a fine-tuning training has been performed for both the Bentham and the Osborne datasets. Four experiments were conducted on different variants or subsamples of the three aforementioned datasets. These experiments tried to measure both the reliability of the proposed normalization and its capability to generalize the results obtained on the training ground truth. To quantify the latter, we have built our analysis of the results upon two metrics, the *Distance Per Pixel* and the *Structural Similarity Index Measure*.

Concerning our four conducted experiments, and even though most of the word images were properly normalized, we observed that those undergoing a significant text resize during the preprocessing stage were significantly more difficult to normalize for the Pix2Pix model. This suggests that resizing is a limiting factor for the capacity of Pix2Pix architectures regarding this normalization task. However, its removal from the global preprocessing procedure may affect the recognition performance of a subsequent recognition model. On the other hand, some adjustments in the proposed architecture and in the steps of the ground truth normalization could be helpful in order to overcome the effect of resizing.

Once the Pix2Pix model proposed in the present work has been trained, it has been jointly integrated with a recognition architecture in an end-to-end configuration. In doing so, the recognition architecture provides Pix2Pix with feedback to improve the normalization phase. So, this new combined HTR model has improved the overall recognition performance of the equivalent non-normalized model. Furthermore, splitting the model into two components (normalization and recognition) enhances its interpretability. Moreover, this architecture could be used for other border normalization tasks, like the recognition of living organisms’ contours. As future works, we will explore some modifications to the Pix2Pix architecture in order to resolve the detected resizing problems, and we will also try to apply this architecture to other similar problems.

## Figures and Tables

**Figure 1 sensors-24-03892-f001:**
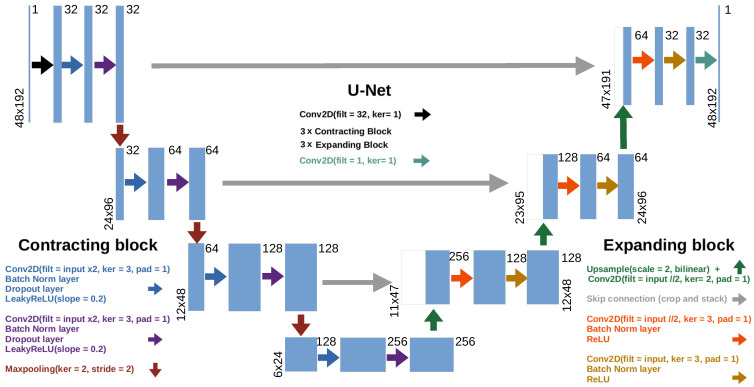
Our U-Net architecture. Each blue box represents a multi-channel feature map. Different arrows represent different operations. The number of channels is shown on top of each box, whereas height and width sizes are specified at the left or right lower edges of the box. White boxes are the cropped feature maps of the encoder part that will be used as skip connections.

**Figure 2 sensors-24-03892-f002:**
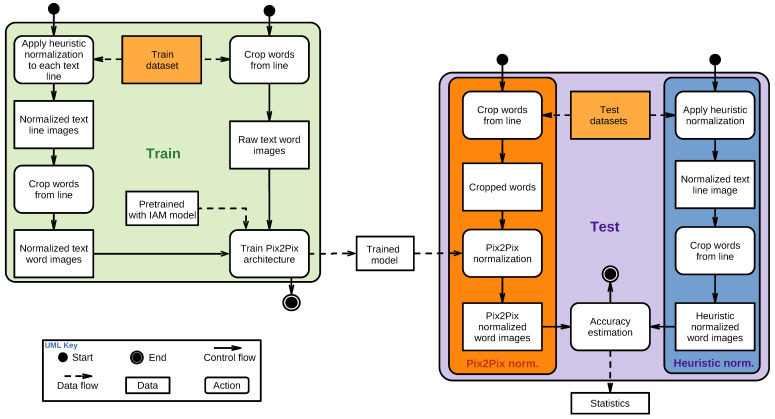
Workflow followed in the different experiments.

**Figure 3 sensors-24-03892-f003:**
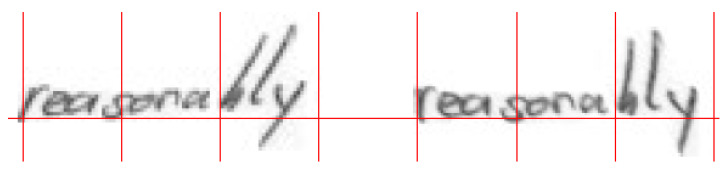
Example of the ad hoc normalization dataset: left word belongs to the *raw IAM dataset*, whereas the right word displays the same image after being manually normalized by visual inspection.

**Figure 4 sensors-24-03892-f004:**
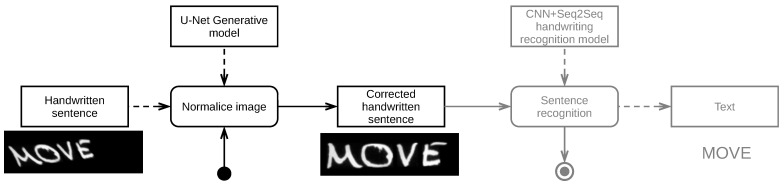
Preliminary end-to-end architecture used in experiment 4.

**Figure 5 sensors-24-03892-f005:**
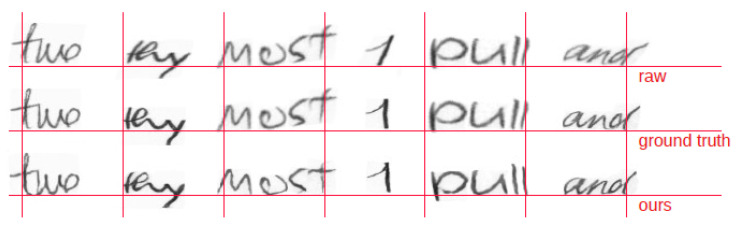
Examples of validation word images. From top to bottom: Raw, ground truth, and generated images by our Pix2Pix architecture with $\lambda_{recon}=5000$.

**Figure 6 sensors-24-03892-f006:**
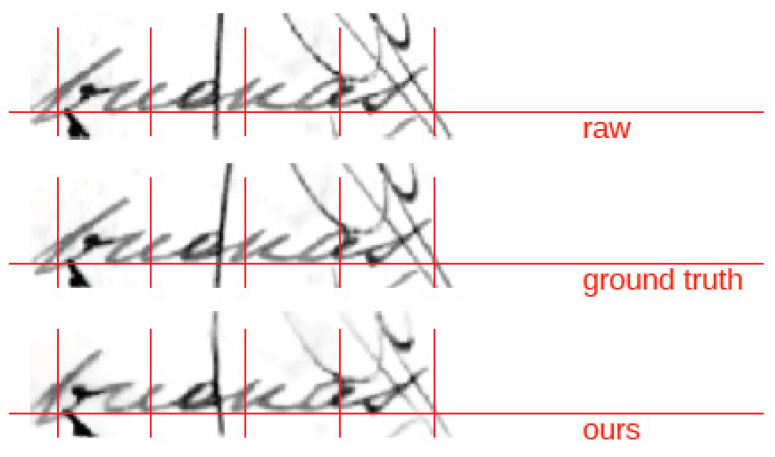
Example of a Spanish handwritten word image from the Osborne dataset that presents overlapping of the handwritten text strokes.

**Table 1 sensors-24-03892-t001:** DPP and SSIM results for the proposed normalization after being applied on the test partitions of the three different datasets. In both cases, the metrics have been computed by comparing each method with the image resulting from applying that very same procedure on each raw test set.

Dataset	Normalization	DPP	SSIM
	Pix2Pix $\left(\lambda_{L1}=50\right)$	0.052	0.700
IAM	Pix2Pix $\left(\lambda_{L1}=500\right)$	0.047	0.718
	Pix2Pix $\left(\lambda_{L1}=5000\right)$	**0.042**	**0.740**
	Pix2Pix $\left(\lambda_{L1}=50\right)$	0.051	0.470
Bentham	Pix2Pix $\left(\lambda_{L1}=500\right)$	0.048	0.498
	Pix2Pix $\left(\lambda_{L1}=5000\right)$	**0.045**	**0.529**
	Pix2Pix $\left(\lambda_{L1}=50\right)$	0.051	0.181
Osborne	Pix2Pix $\left(\lambda_{L1}=500\right)$	0.043	0.282
	Pix2Pix $\left(\lambda_{L1}=5000\right)$	**0.027**	**0.613**

**Table 2 sensors-24-03892-t002:** DPP and SSIM results for the proposed normalization after being applied on the *distorted datasets*. In both cases, the metrics have been computed by comparing each method with the image resulting from applying that very same procedure on each raw (non-distorted) dataset.

Dataset	Normalization	DPP	SSIM
Distorted IAM	Pix2Pix $\left(\lambda_{L1}=50\right)$	0.036	0.758
	Pix2Pix $\left(\lambda_{L1}=500\right)$	0.031	0.784
	Pix2Pix $\left(\lambda_{L1}=5000\right)$	**0.016**	**0.862**
	Heuristic method [14]	0.046	0.772
Distorted Bentham	Pix2Pix $\left(\lambda_{L1}=50\right)$	**0.032**	0.737
	Pix2Pix $\left(\lambda_{L1}=500\right)$	0.033	**0.747**
	Pix2Pix $\left(\lambda_{L1}=5000\right)$	**0.032**	0.738
	Heuristic method [14]	0.049	0.643
Distorted Osborne	Pix2Pix $\left(\lambda_{L1}=50\right)$	0.025	0.854
	Pix2Pix $\left(\lambda_{L1}=500\right)$	0.025	**0.858**
	Pix2Pix $\left(\lambda_{L1}=5000\right)$	**0.023**	**0.858**
	Heuristic method [14]	0.030	0.806

**Table 3 sensors-24-03892-t003:** DPP, SSIM, and slant-angle metrics obtained using as ground truth a manual normalization of 100 random images from the *raw IAM dataset*.

Normalization	DPP	SSIM
Pix2Pix $\left(\lambda_{L1}=50\right)$	0.089	0.556
Pix2Pix $\left(\lambda_{L1}=500\right)$	0.081	0.582
Pix2Pix $\left(\lambda_{L1}=5000\right)$	**0.073**	**0.595**
Heuristic method [14]	0.102	0.545

**Table 4 sensors-24-03892-t004:** End-to-end preliminary recognition results when using a CNN+Seq2Seq LSTM-based attention model [14] after the normalization.

Normalization	WER	CER
Pix2Pix $\left(\lambda_{L1}=5000\right)$	**18.1%**	**5.7%**
Heuristic method [14]	23.8%	8.8%

## Data Availability

All the algorithms explained in this paper, as well as the manually normalized 100 images from the IAM test dataset used in experiment 3, can be found in https://github.com/jfvelezserrano/Handwritten_Analysis/tree/main/sources/normalization (accessed on 14 June 2024).

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
