# Peer review of "A Pix2Pix Architecture for Complete Offline Handwritten Text Normalization"

_sensors, 2024, doi:10.3390/s24123892_

Round 1

Reviewer 1 Report

Comments and Suggestions for Authors

1. The paper primarily discusses results based on a single dataset, the IAM Handwriting Database. While the results are promising, the generalizability of the proposed model to other datasets or more diverse handwriting samples is not established. I would recommend adding another dataset such as to validate the model's performance across different datasets and handwriting styles.

2. Choosing U-net might be challenging because it requires exact paired images (input-target pairs). Instead, why not consider using CycleGAN, which does not require paired data? This approach would alleviate the burden of collecting a large dataset of matched pairs and still achieve effective normalization.

3. While Figure 2 presents the "Workflow followed in the different experiments," adding a comprehensive methodology figure would provide a clearer overview of the entire experimental process.

4. The manuscript could benefit from more extensive experimental validation, including a broader range of performance metrics. For example, combining Word Error Rate (WER) with other metrics like Character Error Rate (CER) would offer insights into performance at a more granular level, measuring errors at the character level rather than the word level.

5. The manuscript lacks a detailed analysis of cases where the model fails or underperforms. Conducting a Confusion Matrix Analysis to break down error types could provide more specific insights into common mistakes or biases in the model's predictions.

6. The paper primarily compares the proposed Pix2Pix model against heuristic normalization methods. Including a comparison with other state-of-the-art deep learning models used in similar tasks would strengthen the claims of superiority and innovation.

Author Response

First of all, we would like to thank you for your review and comments. These have helped us improve our paper.

1. "The paper primarily discusses results based on a single dataset, the IAM Handwriting Database. While the results are promising, the generalizability of the proposed model to other datasets or more diverse handwriting samples is not established. I would recommend adding another dataset such as to validate the model's performance across different datasets and handwriting styles".

Following your recomendations, we have added the Bentham database (in English) and the Osborne database (a multilingual dataset, although primarily in Spanish) to our tests for experiments 1 and 2.

We have also modified experiments 1 and 2 to include the corresponding comparisons with these new datasets. It was interesting to observe that in experiment 1, the best results were consistently obtained for a lambda parameter equal to 5000, which corroborates the conclusions presented in the previous version. Regarding experiment 2, the results also align with the previous findings.

These new results are consistent with those obtained using only the IAM database. Thus, this addition gives us greater confidence in the generalizability of the proposed method.

2. "Choosing U-net might be challenging because it requires exact paired images (input-target pairs). Instead, why not consider using CycleGAN, which does not require paired data? This approach would alleviate the burden of collecting a large dataset of matched pairs and still achieve effective normalization".

Initially, we considered using CycleGAN. However, the literature suggests that this network is better suited for problems where it is important to maintain the shape while changing the texture, such as transforming an image of a horse into a zebra while keeping the shape intact. In our case, we need to alter the shape because our goal is to correct both slant and skew. This is why we believed that U-net and paired image-to-image translation was more appropriate for our specific requirements.

3. "While Figure 2 presents the "Workflow followed in the different experiments," adding a comprehensive methodology figure would provide a clearer overview of the entire experimental process".

We have improved Figure 2 to include the methodological changes resulting from using multiple databases, not just IAM.

Additionally, we have included a new figure (Figure 4) that we believe provides the reader with a clearer overview of the complete normalization and recognition system.

4. "The manuscript could benefit from more extensive experimental validation, including a broader range of performance metrics. For example, combining Word Error Rate (WER) with other metrics like Character Error Rate (CER) would offer insights into performance at a more granular level, measuring errors at the character level rather than the word level".

Thank you for this recommendation. We have added the Character Error Rate (CER) metric to experiment 4 to provide a more granular insight into performance by measuring errors at the character level in addition to the Word Error Rate (WER).

5. "The manuscript lacks a detailed analysis of cases where the model fails or underperforms. Conducting a Confusion Matrix Analysis to break down error types could provide more specific insights into common mistakes or biases in the model's predictions".

We agree with the reviewer that conducting a confusion matrix analysis to identify and categorize errors introduced by the normalization process would be very interesting. Errors such as uncorrected skew or slant, or character deformation, would be valuable to analyze. However, merely comparing our method to the heuristic method does not allow for this level of analysis, as we cannot be certain that the heuristic method has made the right corrections. This is why we introduced experiment 4, in which the network performing the final recognition provides information on the effectiveness of the normalization.

6. The paper primarily compares the proposed Pix2Pix model against heuristic normalization methods. Including a comparison with other state-of-the-art deep learning models used in similar tasks would strengthen the claims of superiority and innovation.

We have expanded the bibliography to analyze how normalization is addressed in other works. In this regard, we have not found other architectures that perform complete normalization as a preliminary step to the recognition stage. As explained in the previous version, most heuristic normalization processes are based on techniques such as pixel counting, pixel projection and so on. This has been reflected in an expanded introduction that also includes some new bibliographic references.

We expect that the proposed changes address all the items you marked as "Must be improved" in the initial review form, specifically regarding providing sufficient background and including relevant references, the research design, the description of the methods, the presentation of the results, and the conclusions. Thank you once again for your valuable feedback and suggestions. We look forward to hearing any further comments you may have.

Reviewer 2 Report

Comments and Suggestions for Authors

The work is interesting and the authors seem to have put their efforts to make the work clear and readable. The handwritten text is a very difficult task and the handwriting subject is a real  challenge in Pattern Recognition.

However, the authors should mention which language was studied as their methodology may not give good results when applied to other languages. Hence, their approach cannot be looked at as a universal one. Moreover, the selected examples are so easy that any other simple approach can give a successful normalisation result without a problem. Therefore, there should be studied such examples that contain letter overlapping which is usually the most problematic concept in handwritten texts in all languages. 

Author Response

First of all, we would like to thank you for your review and comments. These have helped us improve our paper.

The work is interesting and the authors seem to have put their efforts to make the work clear and readable. The handwritten text is a very difficult task and the handwriting subject is a real  challenge in Pattern Recognition.

Thank you for your kind words and recognition of our efforts. We appreciate your acknowledgment of the challenges involved in handwritten text recognition.

1. "However, the authors should mention which language was studied as their methodology may not give good results when applied to other languages. Hence, their approach cannot be looked at as a universal one. Moreover, the selected examples are so easy that any other simple approach can give a successful normalisation result without a problem. Therefore, there should be studied such examples that contain letter overlapping which is usually the most problematic concept in handwritten texts in all languages".

To address this first point, we have incorporated to the experiments the Osborne handwritten dataset. This dataset primarily contains text in Spanish, with a few samples of text in Italian and English. The results obtained with this database are consistent with those from the IAM dataset alone. Additionally, we have included another dataset in our experiments: the Bentham database (in English). Again, the results obtained with this dataset are are consistent with the previous ones.

To address the second point, the Osborne dataset includes characters with overlapping words, which is a commonly problematic scenario in handwritten historical texts. We have included a new figure (Fig. 6) that showcases the results obtained by our proposed algorithm on such an image. Additionally, we have integrated this discussion into subsection 4.2 of our paper.

Finally, to improve the items that you mark as "can be improved" in the in the initial review form, we have made the following changes:

  • To adress with the item "Are the conclusions supported by the results?" We have added two new datasets (Bentham and Osborne). This addition gives us greater confidence in the generalizability of the proposed method.
  • To adress with the item "Are the methods adequately described?" We have improved Figure 2 to include the methodological changes resulting from using multiple databases, not just IAM. Additionally, we have included a new figure (Fig. 4) that we believe provides the reader with a clearer overview of the complete normalization and recognition system.
  • To adress with the item "Are the results clearly presented?" . We have added the Character Error Rate (CER) metric to experiment 4 to provide a more granular insight into performance by measuring errors at the character level in addition to the Word Error Rate (WER). Also, we discuss the results obtained by the proposed algorithm on an image with some overlaping strokes from other word.

Thank you once again for your valuable feedback and suggestions. We look forward to hearing any further comments you may have.

Round 2

Reviewer 2 Report

Comments and Suggestions for Authors

The paper now looks good, thank you.